# Antibiotic Susceptibility Profiles and Resistance Mechanisms to β-Lactams and Polymyxins of *Escherichia coli* from Broilers Raised under Intensive and Extensive Production Systems

**DOI:** 10.3390/microorganisms10102044

**Published:** 2022-10-16

**Authors:** Mariana Ferreira, Célia Leão, Lurdes Clemente, Teresa Albuquerque, Ana Amaro

**Affiliations:** 1Laboratory of Bacteriology and Mycology, National Reference Laboratory of Animal Health, INIAV—National Institute of Agrarian and Veterinary Research, 2780-157 Oeiras, Portugal; 2University of Évora, 7004-516 Évora, Portugal; 3MED—Mediterranean Institute for Agriculture, Environment and Development, 7006-554 Évora, Portugal; 4CIISA—Centre for Interdisciplinary Research in Animal Health, Faculty of Veterinary Science, University of Lisbon, 1300-477 Lisbon, Portugal

**Keywords:** antibiotic resistance, *Escherichia coli*, broiler production systems, ESBL/AmpC β-lactamases

## Abstract

The intensive and extensive broiler production systems imply different veterinary interventions, including the use of antimicrobials. This study aimed to compare the antimicrobial susceptibility profiles of *Escherichia coli* isolated from both systems, characterize resistance mechanisms to β-lactams and polymyxins, and identify genetic elements such as integrons. *E. coli* isolates recovered from broiler cecal samples were assayed for antimicrobial susceptibility through the broth microdilution technique. The molecular characterization of acquired resistance mechanisms to β-lactams and colistin and the detection of integrons was performed by a multiplex PCR. For most antibiotics tested, the prevalence of reduced susceptibility is higher in commensal and extended-spectrum β-lactamases (ESBL)/AmpC producers from broilers raised in the intensive system, compared with those raised under extensive conditions. SHV-12 was the most common ESBL enzyme found in both production systems. Other ESBL variants such as CTX-M-1, CTX-M-55, CTX-M-14, CTX-M-32, CTX-M-9, TEM-52, and plasmid-encoded AmpC enzyme CMY-2 were also present. MCR-1 was identified in a colistin-resistant isolate from broilers raised under the intensive system. This study highlights the differences in *E. coli* antibiotic susceptibility from both production types and emphasizes that a great deal of work remains to decrease consumption and antimicrobial resistance levels.

## 1. Introduction

Antimicrobial resistance (AMR) is a phenomenon that occurs naturally in microbiomes when microorganisms are exposed to the action of antimicrobials. However, the continuous struggle of bacteria for their ecological space allows those with some resistance mechanisms to be selected, survive, multiply and consequently disperse through the microbial communities [1].

The use of antibiotics for the prevention and treatment of diseases in veterinary medicine and animal production allowed for healthier and more productive animals [1,2,3], improved food safety, and promoted economic growth [4]. However, its misuse contributed to the emergence of resistant strains, jeopardizing the antibiotics’ benefits and curative power [5]. Therefore, it is necessary to reduce the unregulated consumption of antibiotics, both in humans and in animals, including food producers entering the food chain, as they may act as potential transmitters of resistant strains [6].

β-lactams (penicillin) are among the most used class of antibiotics worldwide in humans and animals [7]. The predominant resistance mechanism in Gram-negative bacteria is the production of β-lactamases such as cephalosporinases (extended-spectrum β-lactamases (ESBL) and AmpC β-lactamases), which hydrolyze the β-lactam ring and, thus, cause an irreversible blockade of the enzyme that is essential for the cell wall synthesis of these bacteria [8].

In the last decades, ESBL and AmpC-producing strains have emerged globally in animals, increasing the concern of resistant strains being transmitted to humans either through direct contact with animals and the environment [9], or indirectly, by the consumption of contaminated raw or undercooked meat or other foodstuffs [10,11]. The wide spread of ESBL/AmpC-producing *E. coli* strains in the fecal samples of healthy broilers reveals that there may be critical points during industrial broiler production where these strains can be found and selected by other antimicrobials such as tetracyclines, sulfonamides, and fluoroquinolones, contributing to the co-selection of resistance genes by transfer into mobile and mobilizable genetic elements (MGEs), such as plasmids transposons, and integrons [12,13]. Thus, the joint dissemination of antimicrobial resistance determinants of different antibiotic classes, as well as virulence and biocide resistance genes, are favored [14,15].

Within Europe and in most countries, the occurrence of resistance to third-generation cephalosporins in commensal *E. coli* isolates is generally low, although moderate to high levels of resistance from broilers have been observed in some countries [16]. In Portugal, food-producing animals have been described as sources of ESBL-producing *E. coli*, which is mostly associated with the spread of epidemic plasmids within and among different farms and in poultry [17,18,19]. Enzymes belonging to the CTX-M family (-1; -14, -32) have been described, although SHV-12 seems to be the most common in poultry [20,21,22,23].

Following the original report of plasmid-mediated colistin resistance in China in 2015 [24], several studies in different countries reported a worldwide distribution of the *mcr*-1 gene in *Enterobacteriaceae* isolates from humans, food-producing and companion animals, food, and the environment [25,26,27]. More recently, new gene variants (*mcr*-2 to *mcr*-10) have been identified [28], although *mcr*-1 is the most frequently reported in strains of Gram-negative bacteria including *E. coli* [25,28]. However, in the Northern European countries (Norway, Sweden, and Denmark), colistin resistance and the prevalence of *mcr* genes remain low in poultry, as colistin has never been used regularly in veterinary practice. On the other hand, in southern countries, including France, the prevalence of the *mcr*-1 gene is higher, especially in strains from turkeys compared to those from broilers [27,29].

In the last few years, international action plans to reduce the selection and spread of resistance in the bacterial populations of animals [3] are also prompting poultry producers to adopt alternatives to industrial intensive production systems [6]. Moreover, consumers are also making more careful choices for safer and healthier products.

In this study, we sampled broiler birds grown in two different raising conditions, the extensive and the intensive systems. In the extensive production system, birds are from slow-growth breeds, kept in an open yard during the daytime and confined to a house at night. In the intensive method, birds are from selected fast-growth breeds and kept confined throughout the day and night in climate-controlled housing. Feed, feeding habits, and management rules are also different in both systems. Intensive broiler farming is one of the production methods in which antibiotics are used more and is expected to rise in the coming years [4]. Thus, the present work aimed to characterize the antimicrobial susceptibility profiles of *E. coli* isolates from broilers raised under intensive and extensive production systems in Portugal and the molecular mechanisms of acquired resistance to β-lactams and polymyxins. With this study, we sought the existence of differences in antibiotic resistance in the *E. coli* population colonizing the intestine of broilers from both production systems that will enter the food chain.

## 2. Materials and Methods

### 2.1. Sample Collection and Bacterial Isolates

Broiler cecal samples were collected at slaughter under the scope of monitoring and reporting antimicrobial resistance in zoonotic and commensal bacteria, according to DC2013/652EU [30]. Laboratory procedures for isolating and identifying commensal *E. coli* and extended-spectrum β-lactamase (ESBL/AmpC) *E. coli*-producers followed the protocols defined by the EURL-AR [31]. Briefly, one gram of the cecal contents of five birds per flock was homogenized in 9 mL of buffered peptone water (BPW) (Oxoid, Basingstoke, UK), followed by incubation at 37 °C ± 1 °C for 18–22 h. The enriched samples were plated onto MacConkey Agar (Oxoid, Basingstoke, UK) supplemented with 1 mg/L of cefotaxime (CTX) (Glentham, Corsham, UK), followed by incubation at 44 °C ± 0.5 °C to select the growth of presumptive ESBL/AmpC *E. coli* producers (cefotaxime-resistant *E. coli*, abbreviated as CTXr).

Commensal *E. coli* was recovered from non-supplemented MacConkey Agar after incubation at 37 °C ± 1 °C for 18–22 h. Presumptive *E. coli* colonies were selected for biochemical identification on Coli-ID (bioMérieux Marcy-l’Étoile, France), and those confirmed to be *E. coli* were subcultured and preserved at –80 °C for further tests. A total of 193 *E. coli* isolates (grouped in two populations, commensal and CTXr, according to the isolation procedure) were selected for antimicrobial susceptibility testing: 112 from broilers raised under intensive and 81 from extensive production systems in 2014, 2018, and 2020.

### 2.2. Antimicrobial Susceptibility Testing

Antimicrobial susceptibility was assayed using the broth microdilution technique and commercial EUVSEC microplates (Sensititre^®^, Trek Diagnostic Systems, East Grinstead, UK), by determining the Minimum Inhibitory Concentration (MIC). All isolates resistant to third generation cephalosporins (cefotaxime and ceftazidime) were also assayed on the EUVSEC2 microplate (Sensititre^®^, Trek Diagnostic Systems, East Grinstead, UK) to determine the presumptive phenotype of ESBL/AmpC and carbapenemase producers [16]. Presumptive ESBL *E. coli* producers showed resistance to third-generation cephalosporins (cefotaxime and ceftazidime) and synergy with clavulanic acid (CLA), whereas presumptive AmpC *E. coli* producers showed resistance to second- and third-generation cephalosporins and no synergy with clavulanic acid (CLA).

The following antibiotics are included in the EUVSEC and EUVSEC2 microplates: Ampicillin (AMP); Azithromycin (AZT); Cefotaxime (CTX); Ceftazidime (TAZ); Ciprofloxacin (CIP); Colistin (CL); Chloramphenicol (CHL); Gentamicin (GEN); Nalidixic acid (NAL); Sulfamethoxazole (SMX); Tetracycline (TET); Trimethoprim (TMP); Meropenem (MEM); Tigecycline (TGC); Cefepime (CPM); Cefoxitin (FOX); Ertapenem (ERT); Imipenem (IMI); Ceftazidime x Clavulanic acid (TAZ/CLA); Cefotaxime x Clavulanic acid and Temocillin (TRM). *E. coli* ATCC 25922 strain was used as quality control.

The results were interpreted according to the epidemiological cutoff (ECOFFs) [32]. Wild-type populations were here designated susceptible, whereas non-wild type populations were designated resistant (reduced susceptibility). MIC_50_ (MIC value at which ≥50% of the isolates in a test population are inhibited) and MIC_90_ (MIC value at which ≥90% of the strains within a test population are inhibited) were determined for each antibiotic [33]. Isolates were considered multidrug-resistant (MDR) when decreased susceptibility was evidenced to three or more classes of antibiotics.

### 2.3. DNA Extraction

Genomic DNA for PCR amplification was extracted using the boiling method [31]. Briefly, one or two colonies of each strain were washed in 750 µL of sterile H_2_O and centrifuged at 10,000 rpm for 5 min; the supernatant was discarded, and the pellet resuspended in 100 µL of Tris-EDTA buffer, subjected to boiling on a dry block at 98 °C for 10 min, and centrifuged at 20,000× *g* for 5 min; the supernatant containing the DNA molecules was transferred to a new microtube and stored at −20 °C.

DNA from the isolates selected for Whole Genome Sequencing (WGS) was extracted using the PureLink Genomic DNA kit (Invitrogen, MA, USA), according to the Gram-negative bacteria cell lysis protocol. DNA quantity and purity were checked with a Nanodrop® 2000 spectrophotometer (Thermo Scientific, Waltham, MA, USA) by measuring the absorbance at 260, 280, and 230 nm and the 260/280 and 260/230 ratios. DNA quantity was also assessed with QuantusTM Fluorometer using QuantiFluor^®^ ONE dsDNA Dye kit (Promega, Madison, WI, USA), according to the manufacturer’s recommendations. DNA suspensions were stored at −20 °C.

### 2.4. Identification of Integrons by Polymerase Chain Reaction

The Class I, Class II, and Class III integrons were detected based on the presence of gene sequences characteristic for Integrase 1 (IntI1), Integrase 2 (IntI2), and Integrase 3 (IntI3), respectively. The presence of genes encoding the integrases *Int1*, *Int2,* and *Int3* was evaluated by a polymerase chain reaction (PCR) in all isolates (*n* = 193) using the primers and conditions previously described (Appendix A) [34,35].

DNAs from bacterial strains from INIAV and the European Union Reference Laboratory for Antimicrobial Resistance (DTU Food, National Food Institute, Denmark) were used as positive controls in all PCR amplifications. The amplification reactions were run in a Biometra TOne thermocycler (Analytik Jena, Jena, Germany).

PCR products were analyzed by electrophoresis on agarose gel in Tris-Borate-EDTA (TBE) buffer. The products were then visualized and photographed on an ultraviolet transilluminator (BioDoc-It Imaging System, BioRad, Hercules, CA, USA).

### 2.5. Molecular Characterization of Antimicrobial Resistance Genes

In this study, 59 isolates were selected for genomic characterization: 17 from the extensive system and 42 isolates from the intensive production system.

Isolates evidencing a presumptive ESBL and/or AmpC profile were analyzed by multiplex PCR for screening *bla*_CTX-M-Group_ genes (*bla*_CTX-M-Group1_, *bla*_CTX-M-Group2_, *bla*_CTX-M-Group9_, and *bla*_CTX-M-Group8/25_); *bla*_TEM_; *bla*_SHV_; *bla*_OXA_; and *bla*_PMAβ_ (*bla*_ACC_, *bla*_FOX_, *bla*_MOX,_ *bla*_DHA_, *bla*_CIT_, and *bla*_EBC_). The primers and PCR conditions were used as previously described (Appendix A) [36].

The *bla* gene variants and chromosomal AmpC gene mutations were identified using Sanger sequencing of the PCR amplified products (Appendix A). The polymerase with proofreading activity Supreme NZYProof 2x Colorless Master Mix (NZYTech, Lisbon, Portugal) was used for the PCR reactions. PCR products were purified using ExoSAP-IT™ (Applied Biosystems™, Waltham, MA, USA) and subsequently sent to Eurofins Genomics, Germany GmbH, to proceed with sequencing.

The DNA sequences obtained were visualized in ChromasProTM v2.1.8.0 (Technelysium Pty Ltd., South Brisbane QLD 4101, Australia) and analyzed in the blastn of the NCBI [37], in the blastn of the CARD (Comprehensive Antibiotic Resistance Database) [38], and in the ResFinder 4.1 tool of the CGE (Center for Genomic Epidemiology) [39] to identify the variants as described by Bortolaia et al. [40].

Additionally, isolates evidencing decreased susceptibility to colistin were screened for the presence of plasmid-mediated colistin resistance genes (*mcr-1* to *mcr-10*), using primers and conditions previously described [41,42].

### 2.6. Whole Genome Sequencing

Three multidrug-resistant isolates exhibiting phenotypic resistance to critically important antimicrobials (CIA)—two from broilers raised in the extensive production system and one from industrial broilers raised under the intensive production system—were selected for WGS. According to the World Health Organization, CIAs are the sole or one of the limited available therapies for antimicrobial classes to treat serious bacterial infections in humans and life-threatening infections, being of prime importance to preserve the efficacy of such antibacterial agents. Polymyxins, third and fourth-generation cephalosporins, carbapenems, and macrolides are included in this category of antimicrobials [43].

After genomic DNA extraction, DNA libraries and sequencing were performed by Novogene Europe (Cambridge, UK) using Illumina HiSeq sequencing technology (NovaSeq 6000 S2 PE150 XP sequencing mode). The nucleotide sequences were deposited at European Nucleotide Archive with the accession numbers ERS8292187, ERS82922188, and ERS8292189.

The results were analyzed using the bioinformatics tools of the Center for Genomic Epidemiology [39]: *KmerFinder* 3.2, to identify the bacterial species [44]; *MLST* 2.0, to determine the Multilocus Sequence Type [45]; *PlasmidFinder* 2.1, for the identification of plasmids [46]; *ResFinder* 4.1, to identify antibiotic resistance genes [40]; *SeroTypeFinder* 2.0, for serotype identification [47]; *CHTyper* 1.0 to predict the *E. coli Fim*H type and *Fum*C type [48]; *PathogenFinder* 1.1 to predict bacteria’s pathogenicity towards human hosts [49]; and *MobileElementFinder* v1.0.3 to identify mobile genetic elements and their relation to antimicrobial resistance genes and virulence factors [50]. In addition, *ClermonTyping* was used to identify the phylogroup [51].

### 2.7. Statistical Analysis

The Chi-square test was used to evaluate the statistical differences in the prevalence of antibiotic resistance between the isolates of the two production systems. A statistical analysis was performed using the RStudio 1.4 (J. J. Allaire, Boston, MA, USA). A probability value of *p* ≤ 0.05 was considered to indicate statistical significance.

## 3. Results

### 3.1. Phenotypic Characterization of Antimicrobial Resistance

The frequency of resistance (reduced susceptibility, RS), MIC50 and MIC90 values calculated for commensal and presumptive ESBL/AmpC producers (CTXr) *E. coli* isolates recovered from both production types are described in Table 1.

Regarding commensal *E. coli* isolates from both production systems, a statistically significant difference (*p* ≤ 0.05) was observed between the prevalence of reduced susceptibility to several antibiotics, namely nalidixic acid (*p* = 0.030), azithromycin (*p* = 0.045), chloramphenicol (*p* = 0.02) and trimethoprim (*p* = 0.037). Therefore, commensal *E. coli* from the intensive system showed a significantly higher frequency of resistance for quinolones, macrolides, phenicols and dihydrofolate reductase inhibitor than the isolates from the extensive system. For CTXr isolates, the difference is only significant for chloramphenicol (*p* = 0.05) and cefepime (*p* = 0.01). Thus, ESBL\AmpC *E. coli* producers from the intensive system showed a significantly higher frequency of resistance to amphenicol compared to ESB\AmpC isolates from the extensive system.

*E. coli* isolates that were resistant to third-generation cephalosporins and recovered from both types of production also demonstrated a higher prevalence of resistance to azithromycin, chloramphenicol, colistin, tetracycline, sulfamethoxazole, and trimethoprim compared to commensal isolates. High levels of RS to quinolones and fluoroquinolones are here demonstrated by the MIC_90_ values (Table 1).

Of notice, full susceptibility (to all antibiotics tested) was observed in seven strains (8.6%) from the extensive system and four strains (3.6%) from the intensive system. Overall, the MDR profile was identified in 42 (51.9%) and 77 (68.8%) isolates from broilers from the extensive and intensive system, respectively. The different patterns of multidrug resistance observed are shown in Appendix A. Altogether, 45 MDR profiles were observed, and 15 profiles were common to both types of production. Of the remaining 30, the majority were exclusively showed by strains from the intensive system (*n* = 25). The most frequent antibiotic in the MDR profiles was ampicillin (*n* = 42), followed by ciprofloxacin (*n* = 40), sulfamethoxazole (*n* = 39), and nalidixic acid (*n* = 35). Overall, the most prevalent profile, considering the set of all strains studied, was AMP-CIP-NAL-SMX-TET-TMP (12.6%). Nevertheless, the profile AMP-FOT-TAZ-CHL-CIP-NAL-SMX was shown by 13% of *E. coli* isolates from the intensive system, whereas this profile corresponds to only 2.4% of *E. coli* isolates from the extensive system.

### 3.2. Genotypic Characterization of Antimicrobial Resistance Determinants

Several variants of *bla* genes were identified (Table 2 and Table 3). SHV-12 was the most widespread, although other ESBL variants from the CTX-M (-1; -55; -14; -32; -9) and TEM (-52) families, and plasmid-mediated AmpC (pAmpC) enzymes (CMY-2) have also been identified. Although TEM-1 is not an ESBL/AmpC enzyme, it has been detected in co-occurrence.

### 3.3. Mobilizable Genetic Elements

The presence of integrons was more frequent in isolates from intensive systems than in those from the extensive system (Figure 1). The frequency of Class I integrons was higher than Class II in isolates from both production systems, occurring in 46.4% and 29.6% of the isolates recovered from broilers raised under the intensive and the extensive systems, respectively. Class II integrons, although in fewer numbers, were more frequently found in isolates from the extensive system than in the intensive system. Only one isolate from the intensive system was identified carrying solely Class II integrons (Table 3). Class II integrons were commonly associated with Class I, occurring in 9.8% and 4.9% of isolates recovered from broilers produced in the intensive and semi-intensive systems (Figure 1).

### 3.4. Genomic Characterization of Isolates Selected for Whole Genome Sequencing (WGS)

The genetic profiles of three multidrug-resistant isolates to critical antibiotics are detailed in Table 4. The sequenced isolates exhibit resistance determinants confirming the phenotype. They belong to phylogroup A and ST10 and ST744. Several plasmid replicon types were identified, namely IncX4 in the 2fi strain carrying the *mcr-1.1* gene (Table 4). The *MobileElementFinder* v1.0.3 tool predicted this gene being carried by the IncX4 plasmid and the *tetA* and *bla_SHV-12_* genes being carried by Incl.

All isolates showed over 82% probability of being human pathogens. Overall, bacteriocin genes (*cba, cma, cea*); iron-related genes *(*(*iutA, sitA, iroN*); and adherence genes (*iha*) were found. In addition, other virulence genes including *hlyF* (hemolysin F); *astA* (heat-stable enterotoxin 1); *hra* (heat-resistant agglutinin); *iss* (increased serum survival); *iucC* (aerobactin synthetase); *lpfA* (long polar fimbriae); *mchF* (ABC transporter protein MchF); and *traT* (outer-membrane protein complement resistance), were also noticed. Efflux pumps encoding genes *mdfA* were identified in the three strains, and *marA* only in the strain recovered from the broiler raised under the intensive system.

## 4. Discussion

The antimicrobial resistance of commensal and ESBL/AmpC-producing *E. coli* isolates from broilers raised under two types of production systems are compared in this study, with a special focus on the characterization of ESBL producers.

In commensal *E. coli* isolates from free-range broilers, there is a large discrepancy between MIC_50_ and MIC_90_ values for tetracycline, sulfamethoxazole, and trimethoprim, indicating the existence of two distinct bacterial subpopulations with different levels of resistance [33]—one is wildtype and the other is non-wildtype—colonizing the broilers’ gastrointestinal tract. Although the difference is also large for ciprofloxacin, both values fall within the non-wild-type range, according to EUCAST epidemiological cut-off values. Regarding the isolates from intensive production, the difference is higher for azithromycin, chloramphenicol, and trimethoprim. In fact, it is towards these antibiotics that the prevalence of reduced susceptibility between the isolates from both production systems exhibited a statistically significant difference (*p* ≤ 0.05).

Broilers under intensive growth regimes are more subjected to the onset of infectious disorders, and consequently, to a greater use of antimicrobials and the dissemination of AMR [15]. Antibiotics are usually administered to poultry for treating diseases or for prevention by metaphylactic administration [52]. Their use in low doses as growth promoters have been prohibited in the European Union since 2006 [2]. However, antimicrobial substances are essential in the treatment of intestinal infections such as colibacillosis, necrotic enteritis, and other diseases often caused by *Salmonella* sp., *E. coli*, or *Clostridium* spp., which represent huge economic losses for the industry [52]. *E. coli* strains fluctuate according to the chicken’s age, with globally more resistant strains in younger birds, reinforcing the idea of protecting chicks at the different steps of production, beginning with biosecurity in the parent flock, followed by adequate hatching and transport conditions, and biosecurity measures during the chick’s first days of life [53].

Commensal *E. coli* from both production systems showed a low frequency of resistance to third-generation cephalosporins. Some studies refer that the prevalence of resistance to third-generation cephalosporins observed in commensal *E. coli* isolates is low in broilers, although it is in this animal species that a higher number of *Enterobacteriaceae* resistant to these antimicrobials occur [54,55]. This is in line with what has been described in previous studies carried out in Portugal in poultry cecal samples at slaughter and broiler carcasses, and in other European countries [21,52,56,57].

In this study, the most common phenotype in strains with reduced susceptibility to third-generation cephalosporins is the ESBL found in 82.4% of isolates from the extensive system and 90.5% from the intensive system. Several ESBL variants were identified, SHV-12 being the most widespread, which agrees with studies carried out in Portugal and other European countries [21,22,57]. Nevertheless, other ESBL variants such as CTX-M-55, CTX-M-1, CTX-M-14, CTX-M-32, CTX-M-9, TEM-52, and pAmpC enzymes (CMY-2) were also found in accordance with other reports [13,18,21,22,57]. Nonetheless, the occurrence of ten CTX-M-55-producing strains was unexpected, as it is an uncommon variant in Europe, but frequent in Southeast Asian and South American countries [58,59,60,61]. We note that, although previously identified in Portugal in pets [62] and retail broiler meat and beef [12], it is here identified in broilers for the first time.

Since the use of third-generation cephalosporins is not allowed in European poultry production, some authors suggest that the emergence of ESBL/AmpC-producing *E. coli* strains is related to the co-selection pressure from the use of antibiotics such as sulfonamides, tetracycline, fluoroquinolones, and other biocide substances such as quaternary ammonium compounds and copper sulfate, whose resistance determinants are carried in MGEs such as transposons, plasmids, and integrons [21,52,56,57,63].

Integrons play a relevant role as genetic reservoirs for the transfer, integration, and dissemination of resistance genes among bacteria [64,65,66]. In this study, the presence of integrons is more frequent in isolates from broilers raised under the intensive system than in the extensive system. Previous studies underlined a higher prevalence of MDR in integron-positive isolates [67,68], which is in line with our study. The Class I integrons proved to be the most prevalent in both types of production, occurring alone or in association with Class II. In addition, Class II was commonly associated with Class I, which is in accordance with other studies [69].

The high levels of reduced susceptibility to quinolones and fluoroquinolones, here demonstrated by the high MIC_90_ values (NAL > 128 mg/L; CIP > 8 mg/L) in both production systems, may be assigned to the great consumption of enrofloxacin in the poultry industry [70].

*E. coli* isolates that are resistant to third-generation cephalosporins and recovered from both types of production also show a higher prevalence of resistance to azithromycin, chloramphenicol, tetracycline, sulfamethoxazole, and trimethoprim compared to commensal isolates, here demonstrated by the high frequency of MDR isolates, 61.5% and 87.8%, from broilers raised under the extensive and intensive production, respectively. With azithromycin not being authorized for use in veterinary practice and animal production in Europe, this reduction in susceptibility can be explained by the cross-resistance with tylosin, an antibacterial of the macrolide class, which is widely used in poultry production to treat respiratory tract infections [52,70].

Our results show that, although in extensive broiler farms antibiotic use is presumed banned or reduced, antibiotic resistance genes (ARG) were present in the bacterial communities isolated from the fecal microbiome and in the environmental samples (not analyzed), as a result. Studies performed on organic farms report that ARGs measured in the farm environment are generally comparable to those found in antibiotic-free farms [71], suggesting that the efficacy of not using antibiotics is limited by the fact that chicks could be colonized by antibiotic-resistant bacteria on their arrival to the farm [72]. The use of other antimicrobials; certain bacterial features such as biofilm formation, transferability, and co-localization with other antimicrobial resistance genes; or heavy metal/biocide tolerance genes in the same genetic platform, may also be relevant [72]. Moreover, the presence of other animals (rodents, arthropods, and wild birds), dust, sludge manure, feed, or surface run-off water, should also be considered important vehicles of antimicrobial resistance determinants [72].

The genome analysis of the sequenced strains showed the presence of resistance determinants to several classes of antimicrobials, including CIA. In the Mediterranean region, tetracyclines, sulfonamides, fluoroquinolones, and polymyxins are the most prescribed classes of antibiotics in the veterinary sector [73]. Of note, strain 2fi carried the *mcr-1* gene, confirming the colistin-resistant phenotype (MIC = 8 mg/L). This gene has already been identified in several bacterial species, from various animal species, the environment, and food products [19,21,27,58,74]. The main cause of the emergence and transmission of the *mcr-1* gene is thought to be the large use of colistin in farm animals [70,75,76]. This isolate is MDR, presenting other resistance genes, including the *bla*_SHV-12_ gene. Several studies reported that the proportion of *mcr-1* genes in *E. coli* bacteria producing ESBL in animals has been increasing considerably, compared to the low prevalence of this gene in strains not producing β-lactamases, suggesting that the use of colistin favors the selection and dissemination of strains carrying resistance genes towards second-, third- and fourth-generation cephalosporins [27,77,78].

Plasmids can acquire new genes through MGEs, making them perfect vectors for disseminating resistance determinants [79,80,81]. In *Enterobacteriaceae*, the plasmids IncF, IncI, IncA/C, IncL, IncN, and IncH carry a greater number of resistance determinants [79,80]. In this study, bioinformatics tools predicted IncI1 carrying *bla*SHV-12 and *tetA* in strains 6fc and 10fc, and IncX4 carrying *mcr-1* in strain 2fi. Several studies report that ESBL/AmpC-encoding genes in poultry, namely the *bla*_SHV-12_ gene, are mainly associated with replicon types IncI1, IncFIB, IncI2, and IncK plasmids [13,57,59,81,82], which is in accordance with our study. Recently, the IncX4 plasmid has been associated with the *mcr-1* and *mcr-2* genes [75,83] and has already been identified in several countries including Portugal [53,84].

The detection of an ST10 MDR ESBL/*mcr-1* strain in poultry is relevant, as this ST belongs to the clonal complex 10 (CC10), which is one of the most frequent found in clinical *E. coli* strains isolated in humans [53,85]. ST744, although not frequent, is sometimes associated with *E. coli* strains causing infections in humans and animals including poultry, and resistant to CIA, namely colistin, fluoroquinolones, and carbapenems [76,86].

The sequenced isolates belong to phylogroup A, which is commonly associated with animal and human commensal *E. coli* strains, being in line with studies performed in Portugal and in some European countries from the Mediterranean region [56,57,73].

## 5. Conclusions

This study confirms that the two production systems show differences in AMR and MGE in *E. coli* strains that colonize the intestine of healthy broilers that will enter the food chain. The prevalence of reduced susceptibility is higher in *E. coli* strains from the intensive system for most antibiotics tested, including CIAs such as fluoroquinolones and macrolides. Moreover, multi-resistant strains from industrial boilers showed more diverse MDR profiles, including several more antimicrobials simultaneously, suggesting that co-resistance to several antibiotics is more frequent in strains from this type of production. Finally, the predominance of Class I integrons found more frequently in industrial chicken isolates highlights the greater potential for horizontal gene transfer in these strains.

In summary, although some differences in the AMR of *E. coli* strains from the extensive and intensive production systems are noteworthy, other measures to mitigate AMR, such as good hygiene practices and biosecurity, should be a priority.

The improvement in management techniques, such as poultry litter treatment and good hygiene practices, may decrease the use of antimicrobials and, thus, the occurrence of MDR strain frequency in poultry farms. These measures can optimize the poultry production sector and increase animal and human health. The urgency of resorting to a holistic, multidisciplinary, and integrative strategy involving the human, animal, and environmental dimensions to take measures to preserve the effectiveness of existing antimicrobials, reduce their inappropriate use, and limit the spread of resistant bacteria is imperative.

## Figures and Tables

**Figure 1 microorganisms-10-02044-f001:**
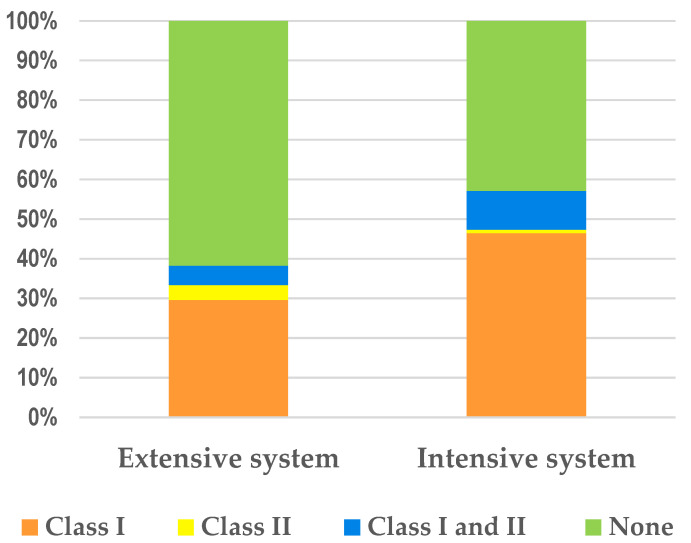
Distribution of Class I and II integrons in *E. coli* strains (*n* = 193) isolated from animals produced in extensive (*n* = 81) and intensive production systems (*n* = 112).

**Table 1 microorganisms-10-02044-t001:** MIC50 (mg/L) and MIC90 (mg/L) values and frequency of reduced susceptibility (%RS) of commensal and presumptive ESBL/AmpC *E. coli* producers (CTXr).

Antimicrobial	Extensive System (*n* = 81)		Intensive System (*n*= 112)		ECOFF (mg/L)
	Commensal (*n* =68)	CTXr (*n* = 13)	Commensal (*n* = 71)	CTXr (*n* = 41)	
β-lactams					
Ampicillin					
MIC50	>64	>64	>64	>64	
MIC90	>64	>64	>64	>64	8
%RS	64.7	100	69	100	
Cefotaxime					
MIC50	≤0.25	16	≤0.25	16	
MIC90	≤0.25	64	≤0.25	>64	0.25
%RS	5.9	100	1.4	100	
Ceftazidime					0.5
MIC50	≤0.5	4	≤0.5	16	
MIC90	≤0.5	32	≤0.5	32	
%RS	5.9	92.3	1.4	100	
Cefoxitin					
MIC50	*	8	*	8	8
MIC90	*	32	*	16	
%RS	0	23.1	0	22	
Meropenem					0.125
MIC50	≤0.03	≤0.03	≤0.03	≤0.03	
MIC90	≤0.03	≤0.03	≤0.03	≤0.03	
%RS	0	0	0	0	
Ertapenem					0.06
MIC50	*	≤0.015	*	≤0.015	
MIC90	*	0.03	*	0.06	
%RS	0	0	0	4.9	
Imipenem					0.5
MIC50	*	0.25	*	0.25	
MIC90	*	0.25	*	0.5	
%RS	0	0	0	2.4	
Quinolones					
Nalidixic Ácid					16
MIC50	>128	>128	>128	>128	
MIC90	>128	>128	>128	>128	
%RS	69.1	76.9	85.9	78	
Ciprofloxacin					0.064
MIC50	0.25	8	2	2	
MIC90	8	>8	>8	>8	
%RS	73.5	76.9	81.7	90.2	
Macrolides					
Azithromycin					16
MIC50	8	8	8	8	
MIC90	8	64	32	32	
%RS	1.5	30.8	11.3	14.6	
Phenicols					
Chloramphenicol					16
MIC50	≤8	≤8	≤8	32	
MIC90	≤8	>128	128	>128	
%RS	4.4	30.8	23.9	65.9	
Polymixins					
Colistin					2
MIC50	≤1	≤1	≤1	≤1	
MIC90	≤1	≤1	≤1	≤1	
%RS	0	0	0	2.4	
Tetracyclines					
Tetracycline					8
MIC50	4	64	64	64	
MIC90	>64	>64	>64	>64	
%RS	50	53.8	56.3	73.2	
Sulfonamides					
Sulfamethoxazole					64
MIC50	≤8	>1024	1024	>1024	
MIC90	>1024	>1024	>1024	>1024	
%RS	36.8	61.5	50.7	87.8	
Dihydrofolate reductase inhibitors					
Trimethoprim					2
MIC50	≤0.25	>32	0.5	0.5	
MIC90	>32	>32	>32	>32	
%RS	27.9	53.8	46.5	34.1	
Glycylcyclines					
Tigecycline					ND
MIC50	≤0.25	≤0.25	0.5	≤0.25	
MIC90	0.5	≤0.25	1	0.5	
%RS	0	0	0	0	
MDR (%)	50	61.5	60.6	87.8	

ECOFF—epidemiological cutoff value; ND—not defined; *—not applicable due to an insufficient number of isolates. MDR—multidrug resistance.

**Table 2 microorganisms-10-02044-t002:** Genotypic characterization of selected *E. coli* strains resistant to third-generation cephalosporins isolated from broilers raised under an extensive production system (*n* = 17).

Sample	Strain	Year	Phenotype	Inhibition by CLA	*bla* Resistance Genes	Integrons (Class)
1fc	Commensal	2014	AMP-FOT-TAZ-CIP-NAL-SMX-TET-FEP	Yes	*bla* _CTX-M-1_	Negative
2fc	Commensal	2014	AMP-FOT-TAZ-CIP-NAL-SMX-TET-TMP-FEP	Yes	*bla*_TEM-1_; *bla*_SHV-12_	I and II
3fc	Commensal	2014	AMP-FOT-TAZ-CIP-NAL-SMX-TET-FEP	Yes	*bla* _CTX-M-1_	Negative
4fc	Commensal	2014	AMP-FOT-TAZ-FEP	Yes	*bla* _SHV-12_	Negative
5fc	CTXr	2018	AMP-FOT-CIP-NAL-FEP	Yes	*bla* _CTX-M-1_	I
6fc *	CTXr	2018	AMP-AZI-FOT-TAZ-CHL-CIP-NAL-SMX-TET-TMP-FEP	Yes	*bla*_CTX-M-14_; *bla*_TEM-1_	I
7fc	CTXr	2018	AMP-FOT-TAZ-FOX	No	*bla*_CMY-2_; *bla*_TEM-1_	Negative
8fc	CTXr	2018	AMP-FOT-TAZ-CIP-NAL-SMX-TMP-FEP	Yes	*bla* _SHV-12_	I and II
9fc	CTXr	2018	AMP-FOT-TAZ-FOX	No	*bla* _CMY-2_	Negative
10fc *	CTXr	2018	AMP-AZI-FOT-TAZ-CHL-CIP-NAL-SMX-TET-TMP-FEP	Yes	*bla*_CTX-M-14_; *bla*_TEM-1B_	I
11fc	CTXr	2018	AMP-FOT-TAZ-FOX	No	*bla* _CMY-2_	Negative
12fc	CTXr	2018	AMP-FOT-TAZ-CIP-NAL-FEP	Yes	*bla*_SHV-12_; *bla*_TEM-type_	Negative
13fc	CTXr	2018	AMP-FOT-TAZ-CIP-NAL-SMX-TET-TMP-FEP	Yes	*bla*_CTX-M-1_; *bla*_TEM-1_	I
14fc	CTXr	2018	AMP-FOT-TAZ-CHL-CIP-NAL-SMX-TET-FEP	Yes	*bla* _SHV-12_	I
15fc	CTXr	2020	AMP-AZI-FOT-TAZ-CIP-NAL-SMX-TET-TMP-FEP	Yes	*bla* _SHV-12_	I and II
16fc	CTXr	2020	AMP-AZI-FOT-TAZ-CHL-CIP-GEN-NAL-SMX-TET-TMP-FEP	Yes	*bla*_CTX-M-55_; *bla*_TEM-type_	I
17fc	CTXr	2020	AMP-FOT-TAZ-CIP-NAL-SMX-TET-TMP-FEP	Yes	*bla* _SHV-12_	I

CLA, Clavulanic Acid; AMP, Ampicillin; AZI, Azithromycin; FOT, Cefotaxime; TAZ, Ceftazidime; CHL, Chloramphenicol; CIP, Ciprofloxacin; GEN, Gentamicin; NAL, Nalidixic Acid; SMX, Sulfamethoxazole; TET, Tetracycline; TMP, Trimethoprim; FEP, Cefepime; FOX, Cefoxitin; *—strain selected for WGS.

**Table 3 microorganisms-10-02044-t003:** Genotypic characterization of *E. coli* strains resistant to third-generation cephalosporins isolated from broilers raised under intensive production system (*n* = 42).

Sample	Strain	Year	Phenotype	Inhibition by CLA	*bla* Resistance Genes	Integrons (Class)
1fi	commensal	2018	AMP-AZI-FOT-TAZ-CHL-CIP-GEN-NAL-SMX-TET-TMP-FEP	Yes	*bla*_CTX-M-55_; *bla*_TEM-type_	I
2fi *	CTXr	2018	AMP-FOT-TAZ-CHL-CIP-COL-NAL-AMX-TET-FEP	Yes	*bla* _SHV-12_	I
3fi	CTXr	2018	AMP-FOT-TAZ-CIP-NAL-SMX-TET-FEP-FOX	Yes	*bla* _TEM-52C_	I
4fi	CTXr	2018	AMP-FOT-TAZ-CIP-NAL-FEP	Yes	*bla* _SHV-12_	I and II
5fi	CTXr	2018	AMP-FOT-TAZ-CIP-NAL-FEP-FOX-ETP	No	*bla*_CMY-2_; *bla*_TEM-type_	Negative
6fi	CTXr	2018	AMP-FOT-TAZ-CHL-SMX-TET-FEP-FOX	Yes	*bla* _SHV-12_	I and II
7fi	CTXr	2018	AMP-FOT-TAZ-CHL-CIP-NAL-SMX-TET-FEP	Yes	*bla* _SHV-12_	I and II
8fi	CTXr	2018	AMP-FOT-TAZ-CHL-CIP-NAL-SMX-TET-FEP	Yes	*bla* _SHV-12_	I
9fi	CTXr	2018	AMP-AZI-FOT-TAZ-CHL-CIP-GEN-NAL-SMX-TET-TMP-FEP	Yes	*bla*_CTX-M-55_; *bla*_TEM-type_	I
10fi	CTXr	2018	AMP-FOT-TAZ-CHL-CIP-NAL-SMX-TET-TMP-FEP	Yes	*bla*_CTX-M-1_; *bla*_TEM-type_	I
11fi	CTXr	2018	AMP-FOT-TAZ-CIP-NAL-SMX-TMP-FEP-FOX	No	*bla* _CMY-2_	I
12fi	CTXr	2018	AMP-FOT-TAZ-CIP-NAL-FEP	Yes	*bla* _SHV-12_	Negative
13fi	CTXr	2018	AMP-FOT-TAZ-CIP-NAL-FEP-IMI	Yes	*bla* _CTX-M-1_	Negative
14fi	CTXr	2018	AMP-FOT-TAZ-CHL-CIP-NAL-SMX-TET-FEP	Yes	*bla* _CTX-M-55_	I
15fi	CTXr	2018	AMP-AZI-FOT-TAZ-CHL-CIP-GEN-NAL-SMX-TET-TMP-FEP	Yes	*bla*_CTX-M-55_; *bla*_TEM-type_	I
16fi	CTXr	2018	AMP-FOT-TAZ-CIP-SMX-FEP	Yes	*bla* _SHV-12_	I
17fi	CTXr	2018	AMP-FOT-TAZ-TET-FEP	Yes	*bla* _CTX-M-1_	I
18fi	CTXr	2018	AMP-FOT-TAZ-CIP-SMX-TMP-FEP-FOX	No	*bla*_CMY-2_; *bla*_TEM-1_	Negative
19fi	CTXr	2018	AMP-FOT-TAZ-CIP-NAL-SMX-TMP-FEP	Yes	*bla* _SHV-12_	II
20fi	CTXr	2018	AMP-FOT-TAZ-CHL-CIP-NAL-SMX-TET-FEP	Yes	*bla* _CTX-M-55_	Negative
21fi	CTXr	2020	AMP-FOT-TAZ-CHL-SMX-TET-FEP	Yes	*bla* _SHV-12_	I
22fi	CTXr	2020	AMP-FOT-TAZ-CHL-CIP-NAL-SMX-TET-FEP	Yes	*bla* _SHV-12_	Negative
23fi	CTXr	2020	AMP-FOT-TAZ-CHL-CIP-NAL-SMX-TET-FEP	Yes	*bla* _SHV-12_	Negative
24fi	CTXr	2020	AMP-FOT-TAZ-CHL-CIP-NAL-SMX-TET-FEP-FOX	Yes	*bla* _CTX-M-55_	I
25fi	CTXr	2020	AMP-FOT-TAZ-CHL-CIP-NAL-SMX-TET-FEP	Yes	*bla* _SHV-12_	I
26fi	CTXr	2020	AMP-FOT-TAZ-CHL-CIP-NAL-SMX-TET-FEP	Yes	*bla*_SHV-12;_ *bla*_TEM-1_	I
27fi	CTXr	2020	AMP-FOT-TAZ-CHL-CIP-NAL-SMX-FEP	Yes	*bla* _SHV-12_	I
28fi	CTXr	2020	AMP-FOT-TAZ-CIP-NAL-SMX-TET-FEP	Yes	*bla* _SHV-12_	I
29fi	CTXr	2020	AMP-AZI-FOT-TAZ-CHL-CIP-GEN-NAL-SMX-TET-TMP-FEP	Yes	*bla*_CTX-M-55_; *bla*_TEM-type_	I
30fi	CTXr	2020	AMP-FOT-TAZ-CIP-NAL-SMX-TMP-FEP-FOX-ETP	Yes	*bla*_CTX-M-32_; mutation on the AmpC promotor	I
31fi	CTXr	2020	AMP-FOT-TAZ-CHL-CIP-NAL-SMX-TET-FEP	Yes	*bla* _SHV-12_	I
32fi	CTXr	2020	AMP-FOT-TAZ-CIP-NAL-SMX-TET-TMP-FEP	Yes	*bla* _CTX-M-9_	Negative
33fi	CTXr	2020	AMP-FOT-TAZ-CIP-NAL-SMX-TET-FEP	Yes	*bla* _CTX-M-1_	Negative
34fi	CTXr	2020	AMP-FOT-TAZ-CHL-CIP-NAL-SMX-TET-FEP	Yes	*bla* _SHV-12_	Negative
35fi	CTXr	2020	AMP-AZI-FOT-TAZ-CHL-CIP-GEN-NAL-SMX-TET-TMP-FEP	Yes	*bla* _CTX-M-55; TEM-1_	I
36fi	CTXr	2020	AMP-FOT-TAZ-CHL-CIP-SMX-TET-FEP	Yes	*bla* _SHV-12_	I
37fi	CTXr	2020	AMP-FOT-TAZ-CHL-GEN-SMX-FEP	Yes	*bla* _CTX-M-32; SHV-12_	I and II
38fi	CTXr	2020	AMP-FOT-TAZ-CHL-CIP-GEN-NAL-SMX-TET-TMP-FEP + FOX	Yes	*bla*_TEM-1_; *bla*_CMY-2_	I and II
39fi	CTXr	2020	AMP-AZI-FOT-TAZ-CHL-CIP-SMX-TET-TMP-FEP	Yes	*bla*_CTX-M-1;_ *bla*_TEM-1_	I and II
40fi	CTXr	2020	AMP-FOT-TAZ-CHL-CIP-SMX-TET-FEP	Yes	*bla* _SHV-12_	I
41fi	CTXr	2020	AMP-AZI-FOT-TAZ-CHL-CIP-GEN-NAL-SMX-TET-TMP-FEP-FOX	Yes	*bla*_CTX-M-55_; *bla*_TEM-type_	I
42fi	CTXr	2020	AMP-FOT-TAZ-CHL-CIP-NAL-SMX-TET-TMP-FEP	Yes	*bla* _SHV-12_	I

CLA, Clavulanic Acid; AMP, Ampicillin; AZI, Azithromycin; FOT, Cefotaxime; TAZ, Ceftazidime; CHL, Chloramphenicol; CIP, Ciprofloxacin; GEN, Gentamicin; NAL, Nalidixic Acid; SMX, Sulfamethoxazole; TET, Tetracycline; TMP, Trimethoprim; FEP, Cefepime; FOX, Cefoxitin; *—strain selected for WGS.

**Table 4 microorganisms-10-02044-t004:** Phenotypic and genomic features of *E. coli* MDR strains selected for WGS (*n* = 3).

Isolate		2fi		6fc		10fc	
Production system		Intensive		Extensive		Extensive	
MLST		ST10		ST744		ST744	
Serotype		O166; H25		O101: H10		O101: H9	
Phylogroup		A		A		A	
fumC/fimH type		fumC11:fimH54		fumC11:fimH54		fumC11:fimH54	
Plasmids		Col; IncB/O/K/Z; IncI1; IncX4; p0111		IncFIB; IncI1; IncW		IncFIB; IncI1; IncI2	
Virulence Genes		*astA, capU, cba, cea, cma, hra, iha, iss, iucC, iutA, sitA, terC, traT*		*cia, cvaC, etsC, etsC, hlyF, iroN, iss, iucC, iutA, mchF, ompT, sitA, terC, traT*		*cma, cvaC, etsC, hlyF, iroN, iss, iucC, iutA, ompT, sitA, terC, traT*	
Pathogenicity		84.1%		86.0%		82.4%	
Antibiotics		MIC (mg/L)	Resistance genes	MIC (mg/L)	Resistance genes	MIC (mg/L)	Resistance genes
Sulfonamides		˃1024	*sul3*	˃1024	*sul1; sul2*	˃1024	*sul1; sul2*
Polymyxins:	Colistin	8	*mcr-1.1*	≤1	-	≤1	-
β-lactams:	Cefotaxime	4	*bla* _SHV-12_	˃64	*bla*_CTX-M-14_; *bla*_TEM-1B_	64	*bla*_CTX-M-14_; *bla*_TEM-1B_
	Ceftazidime	8	*bla* _SHV-12_	1	*bla*_CTX-M-14_; *bla*_TEM-1B_	1	*bla*_CTX-M-14_; *bla*_TEM-1B_
Macrolides:	Azithromycin	8	-	˃64	*mph(A)*	32	*mph(A)*
Phenicols:	Chloramphenicol	32	*cmlA1*	˃128	*catA1*	˃128	*catA1*
Tetracycline		32	*tet(A)*	˃64	*tet(B)*	˃64	*tet(B)*
Trimethoprim		≤0.25	-	˃32	*dfrA17; dfrA5*	˃32	*dfrA17*
Aminoglycosides			*aadA1; aadA2b*		*aadA13; aadA5; aph(6)-Id; aph(3″)-Ib; aph(3′)-Ia;*		*aadA5; aph(6)-Id; aph(3″)-Ib; aph(3′)-Ia;*
Biocides			*qacE*		*qacEdelta1*		*qacEdelta1*
Efflux pumps			*mdf(A); marA*		*mdf(A)*		*mdf(A)*

## Data Availability

The data that support the findings of this study are available within the article. Whole genome sequence data were deposited at European Nucleotide Archive with the accession numbers ERS8292187, ERS82922188, and ERS8292189. Preliminary data were presented as a poster entitled: Antibiotic resistance profiles of *Escherichia coli* isolated from broilers raised under intensive and semi-intensive production systems in the 3rd International Conference of the European College of Veterinary Microbiology, 16–17 October 2021.

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
