# Peer review of "Antibiotic Susceptibility Profiles and Resistance Mechanisms to β-Lactams and Polymyxins of Escherichia coli from Broilers Raised under Intensive and Extensive Production Systems"

_microorganisms, 2022, doi:10.3390/microorganisms10102044_

Round 1
Reviewer 1 Report
This manuscript described the antibiotic susceptibility profiles of E. coli from broilers raised under intensive and extensive production systems. They also performed whole genome sequencing of the three MDR isolates and found genomic characteristics such as ESBL genes and other resistance genes and other typing data. Although the manuscript well-presented and explained their findings with other studies, it lacks the comprehensive interpretation or comparison regarding the intensive and extensive production systems. Also, since this journal ‘Microorganisms’ is not a veterinary or poultry science journal, the authors should make sure the discussion part used proper references and should indicate if the other studies are about human isolates or animal source isolates.
Suggestion: Is there a separate standard for the abbreviation of the antibiotics in your country or EU? If so, please indicate the document. If not, please use the abbreviations recommended by IUPAC-IUB (Biochemical Nomenclature and Related Documents, 1992). You can check the list from the link (https://journals.asm.org/journal/aac/abbreviations).
Line 31: emphases – Do you mean ‘emphasizes’ or ‘emphases on’? Please check it again and revise it if needed.
Introduction: “intensive” and “extensive” production system were not described in the introduction. Since this journal ‘Microorganisms’ is not a veterinary or poultry science journal, those terms should be explained (definition and characteristics), and also the hypothesis (expected AMR differences based on their production system) should be explained (importance of this study or findings from previous studies). Please add a paragraph explaining those.
Line 45-46: Please add a reference or move [5] to the end of the sentence.
Line 49-50: Please specify the target (in animals or human) and add a reference for the statement.
Line 58-64: Please add a reference or move [11,12] (Line:62) to the end of the sentence.
Line 72-73: You may say that SHV-12 seems the most common ‘in Portugal’ with the references but it is hard to tell if it is the most common in general poultry isolates. Please add more references showing SHV-12 from many other countries or change the sentence.
Line 79-81: Please check again if you are indicating specific countries in EU classification and use the term or capitalize them correctly. (For example, Northern -> northern, southern countries -> southern European countries, and France -> including France)
Line 107-121: Please list the antibiotics you used. Also, indicate the abbreviation for the antibiotics (For example, Ampicillin (AMP)…).
Line 122-123: Please delete the unnecessary lines.
Line 110: How did you define ‘commensal’ isolates? Did you use the term to indicate the isolates susceptible to third generation cephalosporins? Please define the cepha-susceptible group with a different term not to confuse the readers. There are many reports about commensal antibiotic-resistant E. coli.
Line 115-121: Reduced susceptibility (RS) should be explained in this part.
Line 116: was here -> were
Line 125-130: Does the boiling method is used as a standard method for DNA extraction in national labs in EU? If it’s also included in standard methods such as EUCAST or EURL protocol, please indicate that or add a reference for that method.
Line 142: What polymerase did you use for the PCR? Please give the product information in the part.
Line 146: Please add manufacturer location (Biometra Tone, Analytik Jena, Germany)
Line 152: Please check the first line indent with other paragraphs.
Line 157: Instead of “elsewhere”, please use any other words indicating the study you used. (For example, “The primers and PCR conditions were used as previously described [33].” or “…described by Dallenne. Et al. (2010)”)
Line 160: Please indicate the company name and location (Supreme NZYProof 2x Colourless Master Mix, NZYTech, Portugal)
Line 161: Add manufacturer information (… Fischer Scientific, USA)
Line 172-184: Please explain how you sequenced the three isolates. Please add a part explaining what type of sequencing method (PacBio, Illumina) you used with the sequencing machine or company information. Also, you need to upload the sequencing data on-line database (GenBank) and accession numbers should be included in the manuscript.
Line 173-174: Critically Important Antimicrobials (CIA) was not defined or explained before. Please briefly explain which antimicrobials are considered as CIA and included in this study so the readers can easily understand.
Line 175: Please indicate that one is from the intensive production system (For example, one from industrial broilers -> one from industrial broilers using intensive production system).
Line 181: Please change it if you want to keep the format consistently: PathogenFinder -> PathogenFinder (italic)
Line 188: Rstudio -> RStudio
Line 189: Please delete ‘difference’.
Line 192: What about the numbers of resistant isolates? Did you find only reduced susceptibility which are intermediate-resistant (I) and not found resistant (R)? Even if you found only small number of resistant isolates, this will be valuable information. Please add the number of resistant isolates in the text and Table 1 if possible.
Table 1: Since this manuscript is more focused on the ESBL/AmpC, it will be more informative if you separate or indicate the beta-lactam class drugs with others in the table.
Ampicilin -> Ampicillin
Line 206 (Table 1): Please add the meaning of ECOFF at the footnote.
Line 207-211: Instead of saying ‘different’, please indicate which showed ‘higher’ (or lower) resistance.
Line 213: demonstrate -> demonstrated
Line 217-218: Please keep the consistency in writing. (seven strains (8.6%)… four (3.6%) strains -> seven strains (8.6%) … four strains (3.6%) …
Line 224: is -> was
Line 227: is showed -> was shown
Line 230: Please delete the space (line).
Line 238: is -> was
Line 244: Tables -> Table
Figure 1: Since the extensive system has been explained first throughout manuscript, please place the extensive system first (left) in the figure. Also, please check line 265: ‘intensive’ and ‘intensive’ and correct one to ‘extensive’.
(Legend) Please give some spaces between items (I, II, I+II None) or change the text (For example, ‘I’, ‘II’ -> Class I, Class II) for better reading.
Table 2. Please check spaces and typos. (Table 2 – Genotypic -> Table 2. Genotypic)
There is no explanation about the test ‘Inhibition by CA’. Please explain why and how you the test and explain the result with discussion.
(Disucssion) Page 14, Line 26-133: Please add a part explaining AMR of intensive and extensive production systems and add a comprehensive interpretation of your results. Also, please find other studies performed under intensive and extensive production systems if possible.
Page 14, Line 27: Please delete the line if it is not necessary.
Line 28-42: Instead of saying ‘different’, please indicate which showed ‘higher’ (or lower) resistance.
Line 51: intensive -> intensive system
Line 69: You didn’t mention about the semi-intensive system before in the manuscript. Did you use the term for the extensive system, or does it indicate only some of the extensive system in your study? Please clarify the term or change it.
Line 103: carry -> carries or carried
Line 109: report -> reported
Line 117-119: Since you performed WGS, it is not necessary to say ‘not perform plasmid sequencing’. If the complete sequencing is available, please change the sentence.
Line 135-152: Conclusion should be interpretation of the study but only little is mentioned. Since the title is about intensive and extensive production systems, readers will expect the results about the differences in the production systems. Also, it is hard to find discussion on hygiene practice, biosecurity and rearing condition in this study, but the conclusion is about those. Please focus more on the interpretation of the results in this study and re-write the conclusion part.
(Data Availability Statement) Page 16, Line 168-169: Whole genome sequence data should be deposited to the on-line database (GenBank) and the accession numbers should be included in the manuscript. You can check the Instructions for Authors. (https://www.mdpi.com/journal/microorganisms/instructions)
Reviewer 2 Report
Dear Authors
The manuscript “Antibiotic susceptibility profiles of Escherichia coli from broilers raised under intensive and extensive production systems” present interesting results on the antibiotic susceptibility of Escherichia coli isolated in intensive and extensive broiler production. Please review the following comments and suggestions:
Verify the congruence between the title and the objective
The title only mentions antibiotic susceptibility profiles, whereas the objective is to say that the molecular mechanisms of acquired resistance to beta-lactams and polimyxins were also investigated.
Lines 55 and 56: In recent decades, it is mentioned twice in the same statement.
Line 127 and 129: rpm should be quoted as g
Line 141: Add a table with the primers used
Line 146: Add the amplification program used
Line 136 to 143: It should be in the discussion section; conclusions should not have references. Only the last two paragraphs are really the conclusion of the paper (lines 144 to 151).
Kind regards
Round 2
Reviewer 1 Report
This manuscript was revised very well with responses to all comments from the reviewers. There are only minor things to be checked. These may be happened due to the tracking system (not displaying the revised ones), but please make sure they are properly modified.
Please make the numbers with inequality sign(>, <) in consistent format (space between number). (Table 1)
Line 204: Please use the original link of the ResFinder for [39]. If you want to keep using the reference, it [39] can be moved to the end of the sentence (… to identify the variants -> …. to identify the variants as described by XXX et al. [39].)
Line 237: RSstudio -> RStudio
Line 274: strains (3.6%) strains -> strains (3.6%)
Author Response
Dear reviewer,
Thank you for your time revising our work.
We checked and corrected it accordingly to your suggestions.
Best regards